# Deregulated FGF and homeotic gene expression underlies cerebellar vermis hypoplasia in CHARGE syndrome

Tian Yu[1], Linda C Meiners[2], Katrin Danielsen[1†], Monica TY Wong[3], Timothy Bowler[4], Danny Reinberg[5], Peter J Scambler[6], Conny MA van Ravenswaaij-Arts[3], M Albert Basson[1,7]*

[1]Department of Craniofacial Development and Stem Cell Biology, King's College London, London, United Kingdom; [2]Department of Radiology, University Medical Center Groningen, University of Groningen, Groningen, Netherlands; [3]Department of Genetics, University Medical Center Groningen, University of Groningen, Groningen, Netherlands; [4]Department of Internal Medicine, Montefiore Medical Center, New York, United States; [5]Department of Biochemistry and Molecular Pharmacology, Howard Hughes Medical Institute, New York University School of Medicine, New York, United States; [6]Molecular Medicine Unit, University College London Institute of Child Health, London, United Kingdom; [7]MRC Centre for Developmental Neurobiology, King's College London, London, United Kingdom

**Abstract** Mutations in *CHD7* are the major cause of CHARGE syndrome, an autosomal dominant disorder with an estimated prevalence of 1/15,000. We have little understanding of the disruptions in the developmental programme that underpin brain defects associated with this syndrome. Using mouse models, we show that *Chd7* haploinsufficiency results in reduced *Fgf8* expression in the isthmus organiser (IsO), an embryonic signalling centre that directs early cerebellar development. Consistent with this observation, *Chd7* and *Fgf8* loss-of-function alleles interact during cerebellar development. CHD7 associates with *Otx2* and *Gbx2* regulatory elements and altered expression of these homeobox genes implicates CHD7 in the maintenance of cerebellar identity during embryogenesis. Finally, we report cerebellar vermis hypoplasia in 35% of CHARGE syndrome patients with a proven *CHD7* mutation. These observations provide key insights into the molecular aetiology of cerebellar defects in CHARGE syndrome and link reduced FGF signalling to cerebellar vermis hypoplasia in a human syndrome.

*For correspondence: albert.
basson@kcl.ac.uk

†Present address: Neural
Development Unit, University
College London Institute of Child
Health, London, United Kingdom

Reviewing editor: Robb
Krumlauf, Stowers Institute for
Medical Research, United States

## Introduction

The segmental organisation of the embryonic neural tube is imparted by the action of homeobox genes that show defined expression patterns along its anterior-posterior axis, in combination with growth factors secreted from distinct organising centres (*Kiecker and Lumsden, 2012*). The cerebellum is derived from dorsal rhombomere 1 (r1), the anterior-most segment of the embryonic hindbrain. The survival and patterning of r1 is controlled by Fibroblast Growth Factor 8 (FGF8), secreted from the isthmus organiser (IsO), an organising centre located at the boundary between the embryonic midbrain (mesencephalon, mes) and r1 (reviewed by *Nakamura et al., 2005*; *Martinez et al., 2013*). The IsO forms at the expression boundary of two homeobox genes: *Otx2* (Orthodenticle Homeobox 2) in the anterior neural tube and, *Gbx2* (Gastrulation Brain Homeobox 2), in the posterior neural tube. *Fgf8* expression in the IsO is initiated at early (3–5) somite stages in the mouse embryo, resulting in a stable gene-regulatory network at the IsO, where (1) cross-repressive interactions between *Otx2* and *Gbx2* maintain the IsO, (2) *Otx2* represses

**eLife digest** CHARGE syndrome is a rare genetic condition that causes various developmental abnormalities, including heart defects, deafness and neurological defects. In most cases, it is caused by mutations in a human gene called *CHD7*. CHD7 is known to control the expression of other genes during embryonic development, but the molecular mechanisms by which mutations in *CHD7* lead to the neural defects found in CHARGE syndrome are unclear.

During embryonic development, the neural tube—the precursor to the nervous system—is divided into segments, which give rise to different neural structures. The r1 segment, for example, forms the cerebellum, and the secretion of a protein called FGF8 (short for fibroblast growth factor 8) by a nearby structure called the isthmus organiser has an important role in this process. Since a reduction in FGF8 causes defects similar to those found in CHARGE syndrome, Yu et al. decided to investigate if the FGF signalling pathway was involved in this syndrome.

Mice should have two working copies of the *Chd7* gene, and mice that lack one of these suffer from symptoms similar to those of humans with CHARGE syndrome. Yu et al. examined the embryos of these mice and found that the isthmus organiser produced less FGF8. Embryos with no working copies of the gene completely lost the r1 segment. The loss of this segment appeared to be caused by changes in the expression of homeobox genes (the genes that determine the identity of brain segments).

Embryos that did not have any working copies of the *Chd7* gene died early in development, which made further studies impossible. However, embryos that had one working copy of the *Chd7* gene survived, and Yu et al. took advantage of this to study the effects of reduced FGF8 expression on these mice. These experiments showed that mice with just one working copy of the *Fgf8* gene and one working copy of the *Chd7* gene had a small cerebellar vermis. This part of the cerebellum is known to be very sensitive to changes in FGF8 signalling. Yu et al. then used an MRI scanner to look at the cerebellar vermis in patients with CHARGE syndrome, and found that more than half of the patients had abnormal cerebella.

In addition to confirming that studies on mouse embryos can provide insights into human disease, the work of Yu et al. add defects in the cerebellar vermis to the list of developmental abnormalities associated with CHARGE syndrome. The next step will be to test if any mutations in the human FGF8 gene can contribute to cerebellar defects in CHARGE syndrome, and to investigate if any other developmental defects in CHARGE syndrome are associated with abnormal FGF8 levels.

*Fgf8* expression, thus restricting it to r1, and (3) *Fgf8* and *Gbx2* contribute to the maintenance of each other's expression (reviewed by *Joyner et al., 2000*; *Martinez et al., 2013*).

Studies in the mouse embryo have shown that the level of FGF gene expression and signalling from the IsO has to be tightly controlled to ensure normal cerebellar development. Altered FGF signalling in the mes/r1 region preferentially affects the development of the medial cerebellum, the vermis (*Xu et al., 2000*; *Trokovic et al., 2003*; *Basson et al., 2008*; *Yu et al., 2011*), which is derived from precursors in the most anterior part of r1, closest to the source of FGF8 expression (*Sgaier et al., 2005*). The observation that reduced FGF signalling results in hypoplasia of the cerebellar vermis in mice raises the possibility that reduced FGF signalling might underlie vermis hypoplasia in certain human conditions. However, studies in mice have also found that FGF signalling has many essential roles during development and even small reductions in *Fgf8* expression during embryonic development are incompatible with postnatal survival (*Meyers et al., 1998*). These findings suggest that mutations causing sufficiently reduced *Fgf8* expression or signalling throughout the whole embryo to result in cerebellar defects are unlikely to yield viable offspring. Rather, it seems more likely that a disruption of the mechanisms that regulate local *Fgf8* expression at the IsO will be responsible for cerebellar vermis hypoplasia in humans.

CHARGE syndrome (MIM#214800) is an autosomal dominant disorder with an estimated prevalence of 1/15,000. Central nervous system defects have been reported in CHARGE (Coloboma of the eye, Heart defects, Atresia of the choanae, Retarded growth and development, Genital anomalies and Ear malformations or deafness) syndrome (*Lin et al., 1990*; *Tellier et al., 1998*; *Becker et al., 2001*;

*Issekutz et al., 2005*; *Sanlaville et al., 2006*; *Sanlaville and Verloes, 2007*; *Bergman et al., 2011*), including reports of cerebellar defects in pre-term CHARGE fetuses (*Becker et al., 2001*; *Sanlaville et al., 2006*; *Legendre et al., 2012*). Depending on the clinical selection, 60–90% of the individuals suspected for CHARGE syndrome have de novo, heterozygous mutations in the *CHD7* (Chromodomain helicase DNA-binding protein 7, MIM#608892) gene (*Vissers et al., 2004*; *Bilan et al., 2012*; *Janssen et al., 2012*). CHD7 is a member of the SNF2H-like chromatin-remodelling family and has been shown to function as a 'transcriptional rheostat' by maintaining appropriate levels of developmental gene expression (*Schnetz et al., 2010*).

A number of clinical findings led us to hypothesise that some developmental defects in CHARGE syndrome might be caused by insufficient FGF signalling levels. For example, CHARGE syndrome shows significant clinical overlap with 22q11.2 deletion and Kallmann syndromes, both of which have been linked to reduced FGF signalling (*Scambler, 2010*; *Miraoui et al., 2013*; *Corsten-Janssen et al., 2013*; *Randall et al., 2009*). We therefore set out to test the hypothesis that CHD7 is required for normal levels of *Fgf8* expression during development by focusing on the embryonic IsO and cerebellar development.

## Results and discussion

### CHD7 regulates *Fgf8* expression levels in the IsO

We previously reported that mice heterozygous for the $Chd7^{XK403}$ gene-trap allele (henceforth referred to as $Chd7^{+/-}$ mice) phenocopy several aspects of CHARGE syndrome (*Randall et al., 2009*). To determine whether FGF signalling at the IsO was affected by *Chd7* deletion, we first visualised the expression of *Fgf8* in E9.5 embryos by in situ hybridisation. *Fgf8* expression in the IsO appeared reduced in $Chd7^{+/-}$ embryos and was substantially downregulated in $Chd7^{-/-}$ embryos (*Figure 1A–C*). Quantitative RT-PCR analysis confirmed that *Fgf8* transcripts were reduced by 20% in $Chd7^{+/-}$ embryos, and by 40% in $Chd7^{-/-}$ embryos (*Figure 1D*). Furthermore, *Fgf8* expression was reduced by 80% in $Chd7^{+/-};Fgf8^{+/-}$ embryos, compared to 40% reduction in $Chd7^{+/+};Fgf8^{+/-}$ embryos (*Figure 1D*). To ask whether this synergistic genetic interaction between *Chd7* and *Fgf8* loss-of-function alleles translated to defects in FGF signalling, the expression of the FGF target gene *Etv5* was analysed (*Roehl and Nusslein-Volhard, 2001*; *Yu et al., 2011*). Whereas *Etv5* expression was clearly diminished in $Chd7^{-/-}$ embryos compared to wildtype controls (*Figure 1E,G*), it did not appear substantially reduced in $Chd7^{+/-}$ embryos (*Figure 1F*), an observation confirmed by quantitative RT-PCR (*Figure 1H*). However, quantitative analyses showed that *Etv5* expression was reduced by 50% in $Chd7^{+/-};Fgf8^{+/-}$ embryos, compared to wildtype levels in $Chd7^{+/-}$ and $Fgf8^{+/-}$ embryos (*Figure 1H*). These data identified CHD7 as an upstream regulator of *Fgf8* in the IsO and revealed a synergistic relationship between the *Chd7* and *Fgf8* genes.

### Synergistic interactions between *Chd7* and *Fgf8* loss-of-function alleles during development of the cerebellar vermis

Previous studies have shown that the medial cerebellum, the cerebellar vermis, is most sensitive to perturbations in FGF signalling during development (*Broccoli et al., 1999*; *Xu et al., 2000*; *Trokovic et al., 2003*; *Basson et al., 2008*; *Yu et al., 2011*), hence we predicted that *Chd7* deficiency will predispose embryos to cerebellar vermis defects. Cerebellar size was normal in $Chd7^{+/-}$ and $Fgf8^{+/-}$ animals compared to wildtype littermates (*Figure 2A–C*), consistent with the observation that FGF signalling was not substantially reduced in these mutants (*Figure 1H*). To accurately compare the sizes of the cerebellar regions between mice, the volumes of cerebellar hemispheres, paravermis and vermis were calculated from surface area measurements taken from serial sections through postnatal day (P)21 cerebella. This analysis confirmed that cerebellar size was not significantly altered in $Chd7^{+/-}$ or $Fgf8^{+/-}$ mice (*Figure 2E*). Furthermore, cerebellar foliation in the vermis and hemispheres appeared normal in the mutants (*Figure 2A'–C',A"–C"*). As $Chd7^{-/-}$ embryos die by E11.5, cerebellar development could not be analysed in these mutants. However, $Chd7^{+/-};Fgf8^{+/-}$ animals survive and an analysis of cerebellar size revealed a significant reduction in size owing to vermis aplasia (*Figure 2D,D',E*). The cerebellar hemispheres were of normal size (*Figure 2E*) and had normal foliation compared to the controls (*Figure 2D"*). Cerebellar vermis aplasia in $Chd7^{+/-};Fgf8^{+/-}$ animals was already present at birth, confirming that defects arose during embryonic development (*Figure 2F–I,F'–I'*, red asterisk). We also noted that the posterior midbrain (inferior colliculus), another region that is

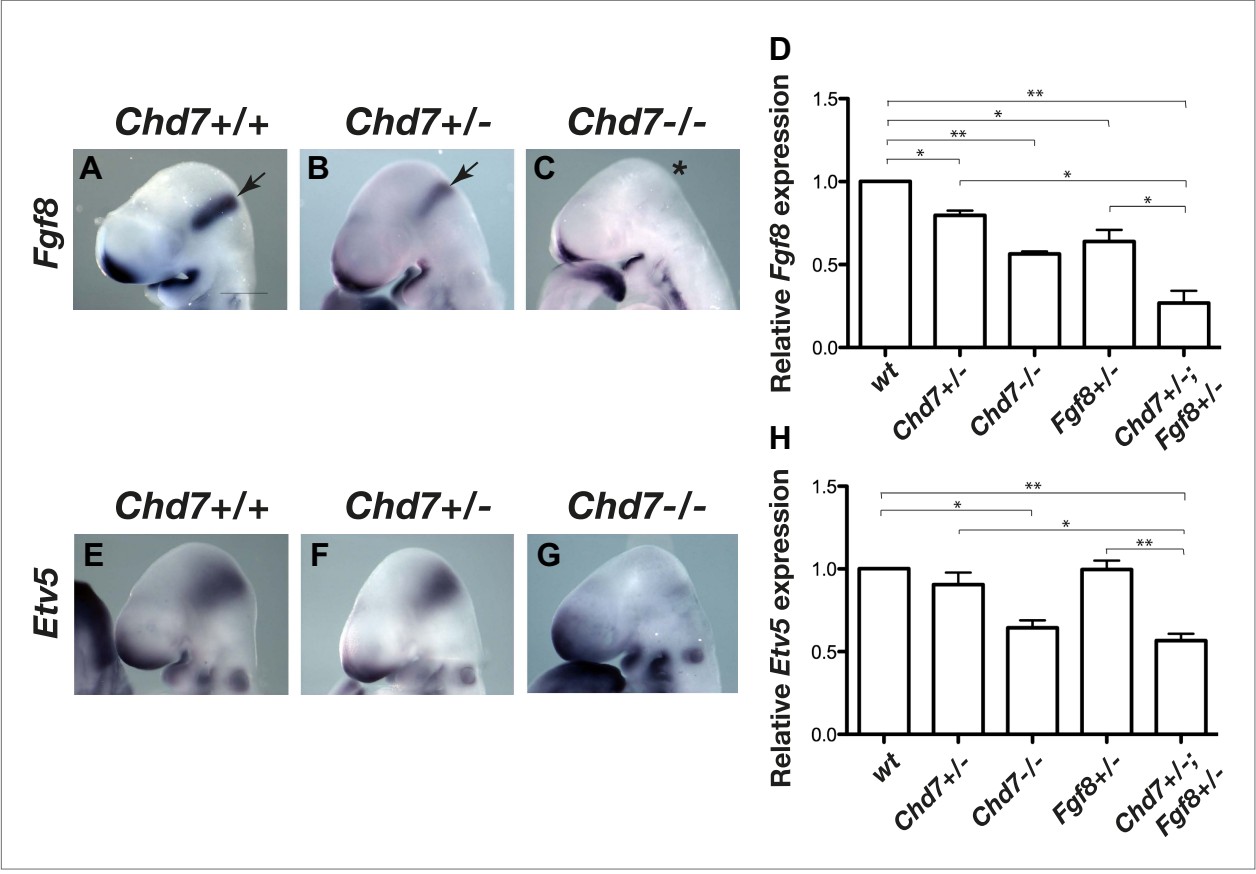

**Figure 1**. Reduced *Fgf8* expression and FGF signalling during early cerebellar development in *Chd7*-deficient embryos. (**A–C**) In situ hybridisation for *Fgf8* at E9.5 shows a *Chd7* gene dosage-dependent reduction in *Fgf8* expression in the mid-hindbrain isthmus organiser (IsO, arrows). Scale bar = 0.5 mm. (**D**) Quantification of *Fgf8* transcript levels in the mes/r1 region of E9.5 embryos. (**E–G**) Expression of the FGF-regulated gene *Etv5* in E9.5 mouse embryos visualised by in situ hybridisation. (**H**) Quantification of *Etv5* gene expression in mes/r1 tissue confirms the in situ hybridisation data and indicates a significant reduction in FGF signalling in *Chd7*<sup>+/−</sup>;*Fgf8*<sup>+/−</sup> and *Chd7*<sup>−/−</sup> embryos. Data represents mean ± standard error of the mean (SEM) from three individual samples for each genotype. *p<0.05, **p<0.001.

particularly sensitive to FGF signalling levels, was abnormal in *Chd7*<sup>+/−</sup>;*Fgf8*<sup>+/−</sup> mutants (**Figure 2I′**, black asterisk). These observations provided functional evidence for a synergistic *Chd7-Fgf8* interaction and indicated that the potential phenotypic consequences of diminished *Fgf8* expression in *Chd7*<sup>+/−</sup> embryos could be revealed by *Fgf8* heterozygosity.

## Deregulated homeobox gene expression and altered r1 identity in the absence of CHD7

The *Chd7* gene encodes a SNF2H-like chromatin remodelling factor that is characterised by the presence of tandem chromodomains in its N-terminal region. Genome-wide chromatin immunoprecipitation studies in cell lines have shown that CHD7 is recruited to distal gene regulatory elements, presumably through interactions between CHD7 chromodomains and methylated lysine 4 residues in histone 3 (H3K4me), present at regulatory elements (*Schnetz et al., 2009, 2010; Engelen et al., 2011*). The *Drosophila* homologue of the *CHD7* subfamily, *kismet*, was identified as a Trithorax gene and *kismet* mutants have reduced expression of homeotic genes and consequent transformations of body segments to more anterior structures (*Daubresse et al., 1999*). We therefore asked whether CHD7 has a role in maintaining the expression of homeobox genes that impart regional identity in the developing neural tube. The homeobox genes *Otx2* and *Gbx2* influence anterior and posterior identity in the developing embryo, respectively, position the IsO and regulate the levels of *Fgf8* expression (*Broccoli et al., 1999; Hidalgo-Sanchez et al., 1999; Millet et al., 1999; Joyner et al., 2000; Heimbucher et al., 2007*). The analysis of *Chd7*<sup>−/−</sup> mouse embryos at E8.25 (4ss), shortly after the initiation of *Fgf8*

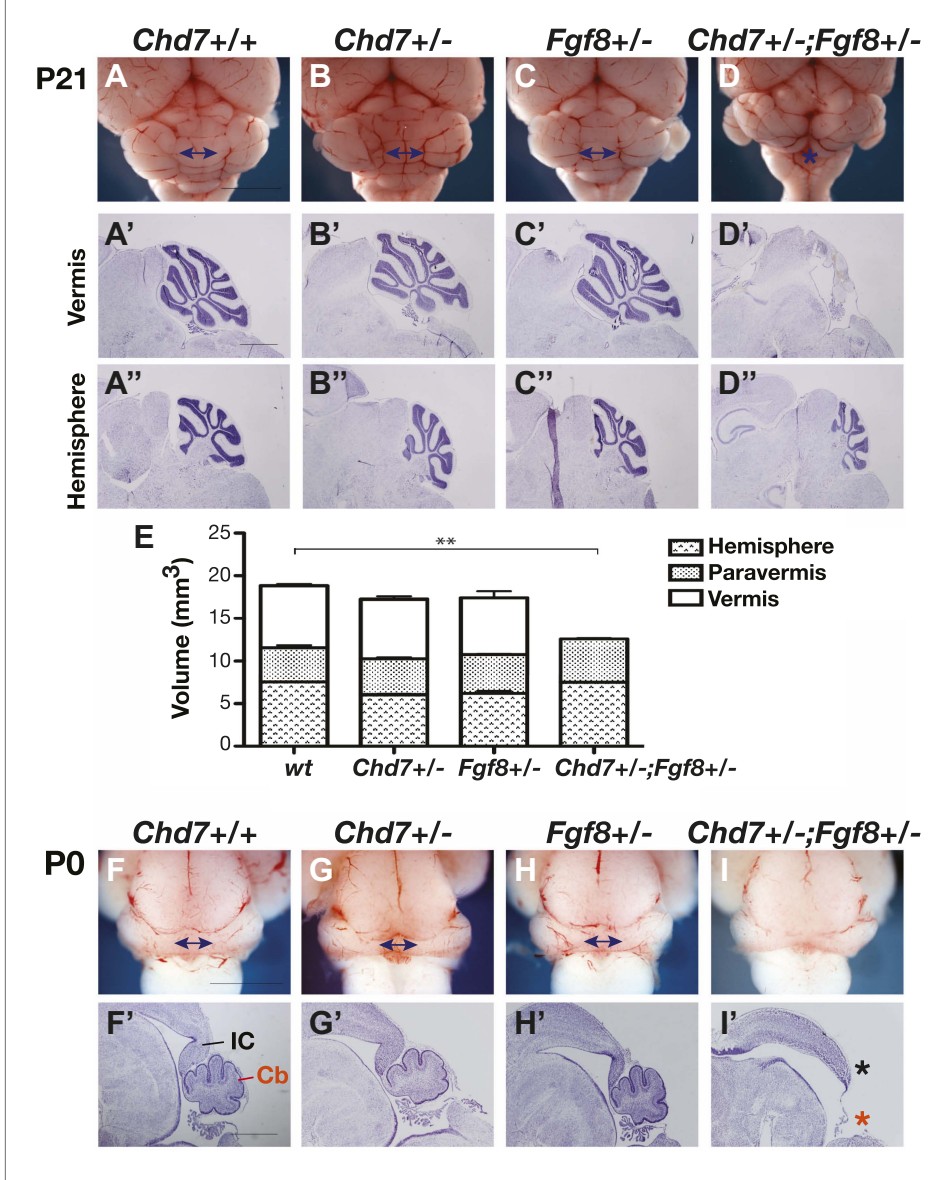

**Figure 2**. *Chd7* and *Fgf8* loss-of-function alleles interact to cause cerebellar vermis aplasia in the mouse. (**A–D**) Wholemount views of the mouse cerebellum at P21. The cerebellar vermis is indicated by a double-headed arrow. *Chd7⁺ᐟ⁻* animals have normal cerebella, indistinguishable from wildtype and *Fgf8⁺ᐟ⁻* control littermates. *Chd7⁺ᐟ⁻;Fgf8⁺ᐟ⁻* animals exhibit vermis aplasia (asterisk in **D**). Scale bar = 5 mm. (A'-D') Cresyl violet-stained sagittal sections through the cerebellar vermis. Note the absence of cerebellar vermis tissue in *Chd7⁺ᐟ⁻;Fgf8⁺ᐟ⁻* embryos (**D'**). (**A"–D"**) Sagittal sections through cerebellar hemispheres. (**E**) Measurements of cerebellar vermis, paravermis and hemisphere sizes in brains from the indicated genotypes. The data represents the mean of three samples with error bars indicating SEM. **p<0.001. (**F–I**) Wholemount views of cerebella at birth (P0), with vermis indicated by arrows. (**F'–I'**) Sagittal sections through P0 brains with inferior colliculus (IC) and cerebellum (Cb) indicated. Note the loss of cerebellar vermis (red asterisk) and abnormal IC (black asterisk) in *Chd7⁺ᐟ⁻;Fgf8⁺ᐟ⁻* animals (**I'**). Scale bar = 1 mm.

expression in the mes/r1 region revealed *Otx2* upregulation and posterior expansion of its expression (*Figure 3A,B*, arrow indicating expanded expression). To investigate how the altered *Otx2* expression domain related spatially to other hindbrain regions, we combined *Otx2* in situ hybridisation with markers to visualise r3+r5 (*Krox20*) and r2 (*Hoxa2*) in the same embryo. Although *Krox20* expression is reduced in *Chd7* mutant embryos (*Alavizadeh et al., 2001*), r3 was still clearly marked, confirming the expansion of *Otx2* expression towards r3 (*Figure 3C,D*). Combined *Otx2/Hoxa2* in situ hybridisation

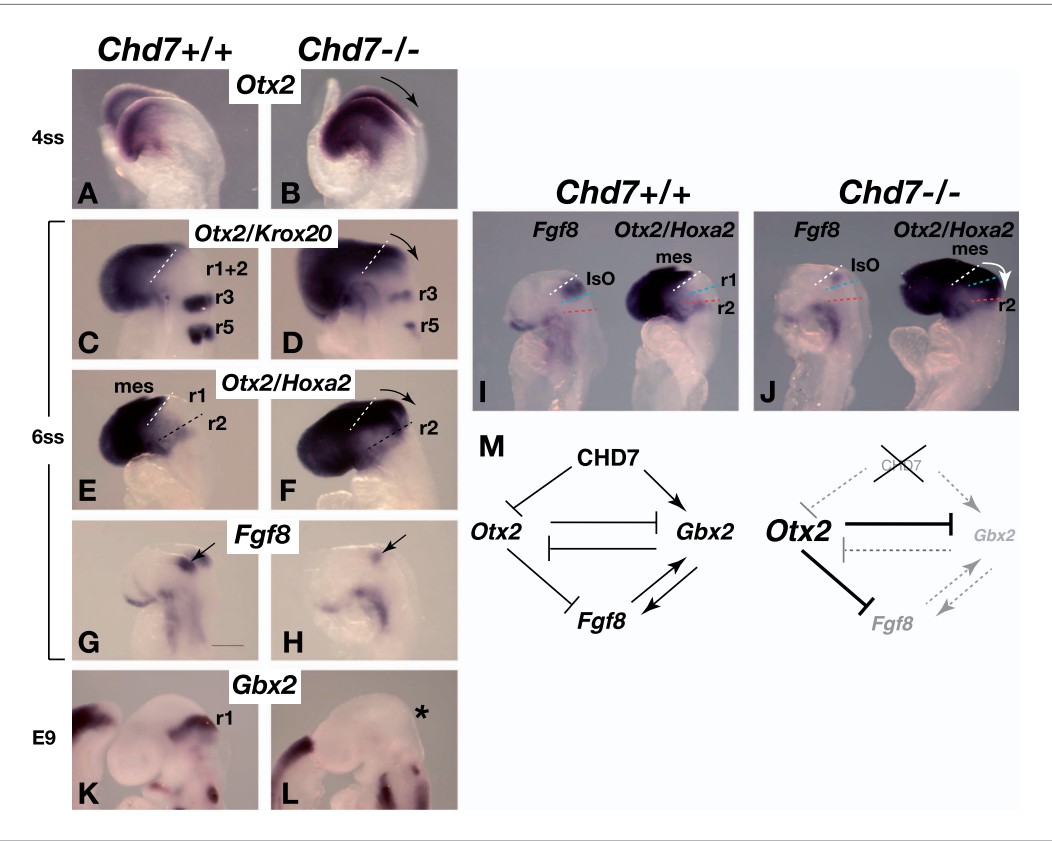

**Figure 3**. *Chd7* loss results in *Otx2* de-repression, loss of rhombomere 1 identity and reduced *Fgf8* expression. (**A** and **B**) In situ hybridisation for *Otx2* in 4 somite stage (ss) embryos. Note the posterior expansion of *Otx2* expression in the mutant embryo (arrow in **B**). (**C** and **D**) In situ hybridisation for *Otx2* and *Krox20* to mark the forebrain/mesencephalon and rhombomeres 3 and 5 (r3 and r5), respectively in 6 ss embryos. Note the posterior expansion of *Otx2* (arrow) towards r3. (**E** and **F**) In situ hybridisation for *Otx2* and *Hoxa2,* to mark the forebrain/ mesencephalon and r2, respectively in 6 ss embryos. Note the posterior expansion of *Otx2* (arrow) and apparent loss of the *Otx2-Hoxa2*-negative r1 in the *Chd7*⁻/⁻ embryo. (**G** and **H**) *Fgf8* in situ hybridisation on 6 ss embryos. Note the initiation of *Fgf8* expression at the correct position in the mutant (**H**), despite posteriorised *Otx2* expression. (**I** and **J**) Side-by-side comparison of *Fgf8* and *Otx2/Hoxa2* expression in 6 ss *Chd7*⁺/⁺ and *Chd7*⁻/⁻ embryos. Note the posterior expansion of *Otx2* expression (white arrow) and downregulated *Fgf8* expression in the *Chd7*⁻/⁻ embryos, compared to wildtype controls. Also note that *Fgf8* expression is initiated at the correct position in the *Chd7*⁻/⁻ embryo, with no evidence of a repositioning of the IsO in response to posterior expansion of *Otx2* at this stage of development. (**K** and **L**) In situ hybridisation for *Gbx2* suggesting the loss of r1 identity by E9. (**M**) Summary of regulatory interactions at the IsO in *Chd7*⁺/⁺ vs *Chd7*⁻/⁻ embryos. The loss of *Otx2* repression and *Gbx2* maintenance by CHD7 are predicted to result in reduced *Fgf8* expression in *Chd7*-deficient embryos. mes = mesencephalon, r1 = rhombomere 1, r2 = rhombomere 2, IsO = isthmus organiser.

experiments suggested that the expansion of *Otx2* expression included most of r1, as indicated by the absence of the *Otx2/Hoxa2*-negative r1 in *Chd7*⁻/⁻ embryos (**Figure 3E,F**).

We further confirmed that the posterior expansion of *Otx2* expression in these early *Chd7*⁻/⁻ embryos was associated with reduced *Fgf8* expression (**Figure 3G,H**), in agreement with previously reported repression of *Fgf8* expression by OTX2 (**Acampora et al., 2001**; **Heimbucher et al., 2007**). A side-by-side comparison of stage-matched embryos indicated that *Fgf8* expression was initiated at the correct position in *Chd7*⁻/⁻ embryos (**Figure 3I,J**). This observation showed that the posterior expansion of *Otx2* expression was not associated with a re-positioning of the IsO and indicated that *Otx2* was mis-expressed in the anterior hindbrain of *Chd7*⁻/⁻ embryos. Therefore, we asked whether the abnormal expansion of *Otx2* expression into the hindbrain just posterior to the IsO affected the identity of r1. Indeed, expression of the homeobox gene *Gbx2*, a marker of r1, was downregulated in

*Chd7*−/− embryos (*Figure 3K,L*). These findings are consistent with the known regulatory interactions between *Otx2*, *Gbx2* and *Fgf8* (*Figure 3M*) (*Broccoli et al., 1999*; *Millet et al., 1999*). We conclude that CHD7 functions as a key regulator of homeobox gene expression in the early neural tube and that the loss of *Chd7* results in the altered expression of *Otx2* and *Gbx2,* and the concomitant transformation of r1 into a more anterior identity. Interestingly, the effect of *Chd7* mutation on *Otx2* expression appears to be highly context-dependent as *Otx2* is reported to be downregulated in the otic and olfactory regions of *Chd7*-deficient embryos (*Hurd et al., 2010*; *Layman et al., 2011*).

## CHD7 is associated with *Otx2* and *Gbx2* regulatory elements

The data presented thus far indicated that CHD7 is required for normal *Otx2* and *Gbx2* gene expression. We therefore sought evidence for CHD7 recruitment to *Otx2* and *Gbx2* regulatory regions. CHD7-associated chromatin was isolated from the mes/r1 region of E9.5 embryos by chromatin immunoprecipitation (ChIP), and genomic DNA fragments (indicated as #1–#10 in *Figure 4A*) quantified by qPCR. Specific CHD7 binding was observed at three *Otx2* enhancer elements identified by Kurokawa et al. (*Kurokawa et al., 2004a*, *2004b*). The FM1 enhancer, located ~71–73 kb upstream (#8 in *Figure 4A*), and the FM2 enhancer (#1 in *Figure 4A*), located ~118 kb downstream of *Otx2* are by themselves sufficient to direct gene expression to the forebrain and midbrain after E9, and deletion of both enhancers together results in a smaller rostral brain and expanded r1 (*Sakurai et al., 2010*). DNA fragments within these Otx2 enhancers (#1, #8) were specifically enriched by CHD7 ChIP (*Figure 4A*). In addition, CHD7 was also present at the AN enhancer (#9 and #10 in *Figure 4A*) that can direct gene expression to the epiblast and anterior neuroectoderm prior to E9.0 (*Kurokawa et al., 2004b*), consistent with the observation that *Otx2* expression was altered in E7.5 embryos (data not shown). We also detected CHD7 association with regions downstream of *Otx2* (#3), and the promotor regions for *Otx2.1* (#6) and *Otx2.2* (#7) transcripts. No specific enrichment was detected at two negative control regions (#2 and #4). These data suggested that CHD7 is recruited to several key *Otx2* regulatory elements in the embryonic mes/r1 region.

A regulatory region 6 kb upstream of zebrafish *Gbx2* capable of driving gene expression in r1, has been described by *Islam et al. (2006)*. ChIP-qPCR experiments identified substantial CHD7 recruitment to a region 5–6.25 kb upstream (#3, #4 and #5 in *Figure 4B*) as well as 3.7 kb upstream of *Gbx2* (#1) in mes/r1 tissue. These observations suggested that CHD7 might regulate *Gbx2* expression in r1 by interacting with *Gbx2* regulatory elements. However, further experiments will be required to test whether these regions do indeed control *Gbx2* expression in mouse r1.

Taken together, our observations support the supposition that homeobox genes represent key CHD7 targets. The mechanisms controlling CHD7 recruitment to regulatory regions and the action whereby CHD7 might affect gene expression in the embryo remain to be elucidated.

## Cerebellar defects of CHARGE syndrome patients

Cerebellar defects have been reported in pre-term CHARGE fetuses (*Becker et al., 2001*; *Sanlaville et al., 2006*; *Legendre et al., 2012*). To determine whether cerebellar defects are a common postnatal feature of CHARGE syndrome, we systematically examined cerebellar structure in a cohort of 20 patients with CHARGE syndrome and mutations in the *CHD7* gene. MRI scans revealed cerebellar defects in 55% (11/20) of these patients (*Figure 5*; *Table 1*). Patients exhibited cerebellar vermis hypoplasia, varying from slight to pronounced hypoplasia (35%, 7/20, *Figure 5B,C*) and an anticlockwise rotated vermis (35%, 7/20, *Figure 5B,C'*). As a consequence of these abnormalities, fluid-filled spaces surrounding the cerebellum appeared larger. Examples of large foramen of Magendi and fourth ventricle (50%, 10/20) and large subcerebellar cistern (25%, 4/20) are indicated in *Figure 5B–D*. Thus, cerebellar defects in CHARGE syndrome have some clinical similarities to Dandy-Walker malformations (vermis hypoplasia and anticlockwise rotated vermis), without the overt posterior fossa enlargement typical of Dandy-Walker malformation (*Doherty et al., 2013*). Two patients with vermis hypoplasia exhibited broad gait or ataxia, consistent with defects that disrupt cerebellar function (*Table 1*). Furthermore, 25% (5/20) of the patients had foliation abnormalities (*Figure 5D,D'*, *Table 1*), implying additional roles for CHD7 during the process of foliation. We conclude that a substantial proportion of patients with CHARGE syndrome present with cerebellar vermis hypoplasia. The incomplete penetrance of cerebellar vermis defects in patients with *CHD7* mutations is consistent with our studies in the mouse, which suggests that mutations in FGF pathway genes are likely to substantially modify the severity of cerebellar defects in CHARGE syndrome.

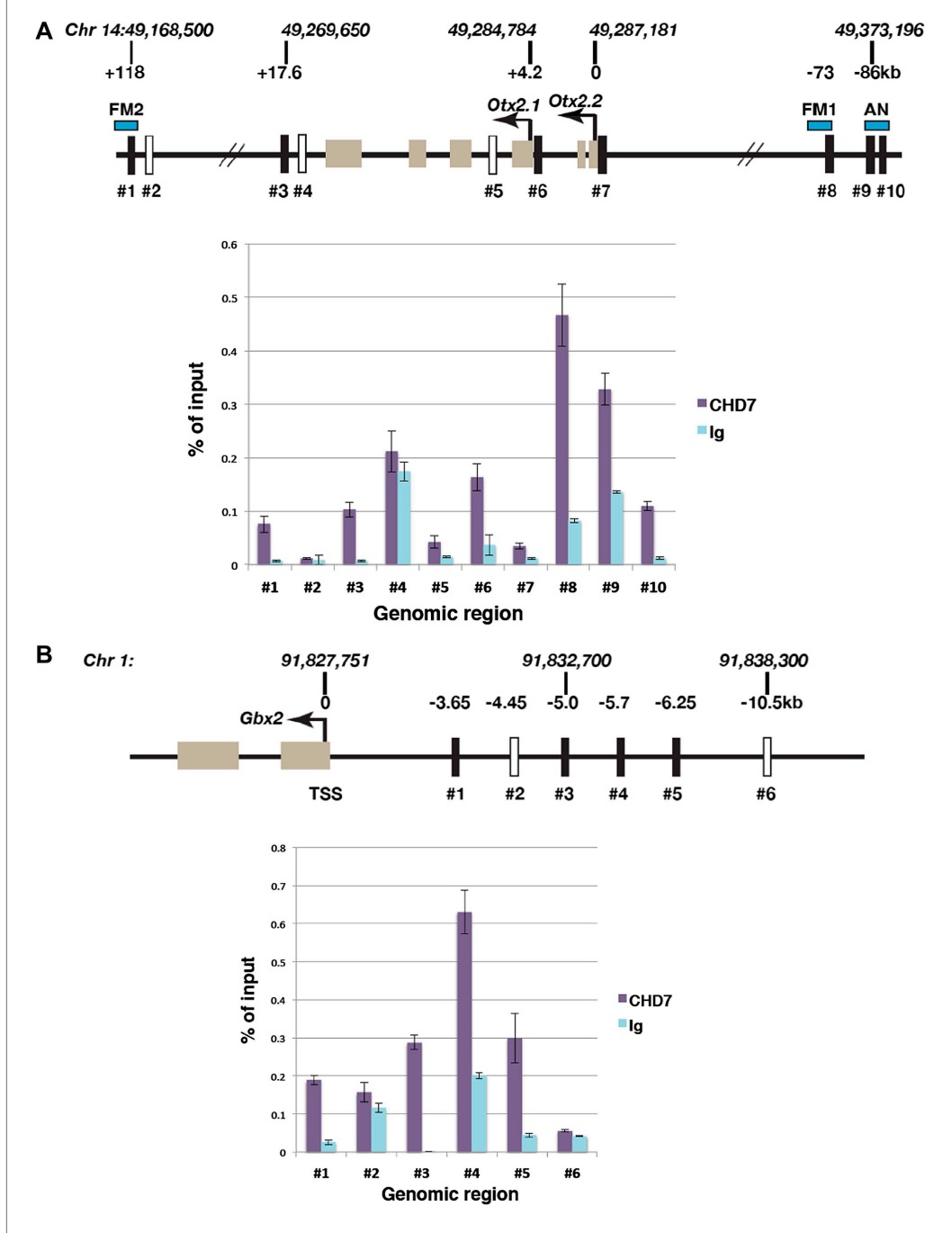

**Figure 4**. Association of CHD7 with *Otx2* and *Gbx2* regulatory regions in the mes/r1 region. (**A**) Genomic map of the mouse *Otx2* locus. The transcriptional start sites of *Otx2.1* and *Otx2.2* transcripts are indicated by arrows and exons by tan-coloured boxes. Positions on chromosome 14 indicated above the diagram are according to the mm9 genome assembly and numbers below the horizontal lines indicate approximate positions relative to the *Otx2.2* transcriptional start site. Known *Otx2* enhancer regions FM1, FM2 and AN are indicated by blue boxes (***Kurokawa et al., 2004a***, ***2004b***). The location of DNA fragments amplified by qPCR after ChIP are indicated by rectangular boxes numbered #1–#10. Open boxes indicate negative control regions. ChIP-qPCR data are presented in a graph, with % of input DNA on the Y-axis and amplified region on the X-axis. Results from ChIP reactions using a CHD7-specific antiserum are in magenta and control Ig in turquoise. Error bars indicate standard deviation from reactions performed in triplicate. (**B**) Genomic map of the mouse *Gbx2* locus with the transcriptional start site (TSS) indicated by an arrow and exons by tan-coloured boxes. Positions on chromosome 1 indicated above the diagram are according to the mm9 genome assembly and numbers below the horizontal lines indicate approximate positions relative to the TSS. The location of DNA fragments amplified by qPCR after ChIP are indicated by rectangular boxes numbered #1–#6. Open boxes indicate negative control regions. ChIP-qPCR data are presented in a graph, with % of input DNA on the Y-axis and amplified region on the X-axis. Results from ChIP reactions using a CHD7-specific antiserum are in magenta and control Ig in turquoise. Error bars indicate standard deviation from reactions performed in triplicate.

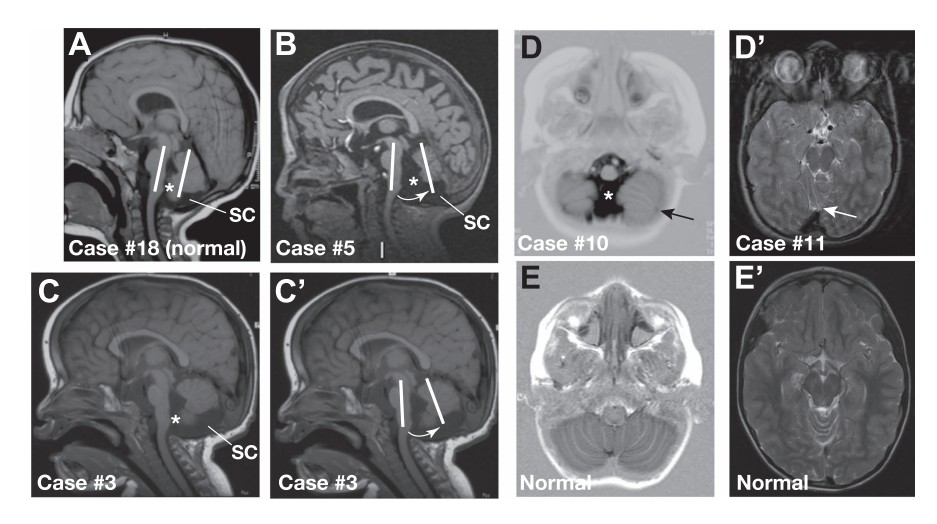

**Figure 5**. Representative sagittal MRI scans of CHARGE syndrome patients. (**A**) Sagittal T1 scan of patient #18 showing a normal vermis with a normal position, foramen of Magendi (asterisk) and subcerebellar cistern (SC). The orientation of the cerebellum relative to the brainstem is indicated by two parallel white lines. (**B**) Sagittal T1 scan of patient #5 showing pronounced vermis hypoplasia with an anticlockwise rotated axis relative to the axis of the brainstem (arrow), and ensuing large foramen of Magendi (asterisk) and subcerebellar cistern (SC). Cerebellar hemispheres are normal (not shown). (**C** and **C'**) Illustrative sagittal T1 MRI images of patient #3 showing a slightly hypoplastic vermis. The white lines and arrow in **C'** indicate the anticlockwise-rotated axis of the vermis compared to the axis of the brainstem, with ensuing large foramen of Magendi (asterisk) and subcerebellar cistern (SC) indicated in **C**. (**D**) Transverse Inversion Recovery MRI image of patient #10 showing abnormal foliation in the caudal cerebellar hemispheres extending into the cerebellar tonsils (arrow). Also note a wide foramen of Magendi (asterisk). (**D'**) Transverse T2 MRI image of patient #11, with abnormal foliation in the anterior vermis indicated by an arrow. (**E**) Transverse Inversion Recovery image and (**E'**) T2 MRI image of a control patient with normal cerebellum.

In summary, this study identifies the chromatin-remodelling factor CHD7 as a key upstream regulator of homeobox gene expression and positional identity in the early neural tube and demonstrates a connection between CHD7 haplo-insufficiency, reduced FGF signalling and cerebellar defects in a human syndrome. We propose that CHD7 remodels chromatin at multiple *Otx2* and *Gbx2* regulatory elements, thereby modifying higher order chromatin architecture and interactions with tissue-specific transcription factors at these loci. Although we cannot completely rule out the possibility that CHD7 also directly fine-tunes *Fgf8* expression in addition to affecting *Otx2* and *Gbx2* expression, the finding that *Fgf8* expression is not substantially changed in the pharyngeal region of *Chd7*[−/−] embryos (compare e.g., **Figure 3G** with H), suggest that such effects will have to be mediated by CHD7 recruitment to tissue-specific *Fgf8* regulatory elements.

Our findings predict that mutations and epigenetic alterations of *OTX2* and *GBX2* regulatory regions are likely to contribute to cerebellar hypoplasia in humans and that *OTX2*, *GBX2* and *FGF8* deregulation might underlie other developmental defects associated with CHARGE syndrome.

## Materials and methods

### Animals

The *Chd7*[XK403] and *Fgf8*[lacZ/+] loss-of-function alleles were maintained on C57BL/6J and C57BL/6J × DBA/2J F1 backgrounds for these studies (*Ilagan et al., 2006*; *Randall et al., 2009*). Tail DNA preparations were genotyped by PCR as described in the original publications. All animal procedures were approved by the UK Home Office.

### Histology and volumetric analysis

Brains were dissected in ice-cold PBS, fixed in 4% paraformaldehyde (PFA) overnight at 4°C, before dehydration and embedding in paraffin wax. Volumetric measurements were carried out on P0 and

**Table 1.** Cerebellar findings on MRI scans

| Patient | Sex; age at MRI (y;m) | Cerebellum | Suggestive neurological features* (age at last examination, y;m) | CHD7 mutation | |
|---|---|---|---|---|---|
| 1 | M (1;1) | Pronounced vermis hypoplasia with anticlockwise rotated axis, large foramen of Magendi and large subcerebellar cistern, fissure vermis | None (1;1) | nonsense | 934C>T |
| 2 | M (0;1) | Slight caudal vermis hypoplasia with slightly anticlockwise rotated axis, abnormal foliation, large foramen of Magendi, normal subcerebellar cistern | Ataxic gait (4;4) | nonsense | 7160C>A |
| 3 | M (1;0) | Slight caudal vermis hypoplasia with anticlockwise rotated axis, large foramen of Magendi, large subcerebellar cistern (*Figure 5C,C'*) | None (12;4) | deletion | 3202-?8994?del |
| 4 | F (0;3) | Slight caudal vermis hypoplasia, with anticlockwise rotated axis, large foramen of Magendi, normal subcerebellar cistern | None (2;2) | frameshift | 7106delT |
| 5 | M (5;7) | Pronounced vermis hypoplasia, with anticlockwise rotated axis, large foramen of Magendi and large subcerebellar cistern (*Figure 5B*) | None (7;10) | frameshift | 4779delT |
| 6 | M (0;1) | Slight caudal vermis hypoplasia, with anticlockwise rotated axis, large foramen of Magendi and large subcerebellar cistern | None (5;2) | frameshift | 5680_5681delAG |
| 7 | F (2;9) | Slight caudal vermis hypoplasia, with slightly anticlockwise rotated axis, large foramen of Magendi and large subcerebellar cistern | Broad gait (11;6) | missense | 3973T>G |
| 8 | M (1;8) | Large foramen of Magendi, large fourth ventricle (only on sagittal scans), normal subcerebellar cistern | None (12;2) | splice site | 5535-7G>A |
| 9 | M (2;2) | Large foramen of Magendi, large fourth ventricle (only on sagittal scans), normal subcerebellar cistern. Abnormal foliation in anterior vermis | None (6;2) | nonsense | 3173T>A |
| 10 | F (1;1) | Abnormal foliation caudal cerebellar hemispheres and tonsils, large foramen of Magendi (*Figure 5D*) | None (13;0) | splice site UV | 3340A>T |
| 11 | F (15;10) | Abnormal foliation in anterior vermis (*Figure 5D'*) | None (18;0) | splice site | 3990-1G>C |
| 12 | M (10;3) | Abnormal foliation in anterior vermis | Motor dyspraxia (16;10) | frameshift | 5564dupC |
| 13 | M (0;1) | Normal (indented cranial pons) | None (0;11) | frameshift | 1820_1821insTTGT |
| 14 | F (15;10) | Normal (large fourth ventricle) | None (20;6) | nonsense | 4015C>T |
| 15 | F (0;1) | Normal, (split caudal vermis) | None (5;9) | nonsense | 7879C>T |
| 16 | M (0;6) | Normal | Broad gait (10;6) | splice site | 2238+1 G>A |
| 17 | M (1;10) | Normal | None (6;4) | nonsense | 1480C>T |
| 18 | F (2;10) | Normal (*Figure 5A*) | None (17;3) | frameshift | 7769delA |
| 19 | M (1;0) | Normal | None (16;9) | nonsense | 1714C>T |
| 20 | M (6;3) | Normal | None (12;10) | splice site | 2443+5 G>C |

*all children show motor delay due to vestibular defects.

P21 cerebella. Serial, sagittal 10 μm sections of the cerebellum were dried overnight at 42°C, rehydrated and stained with 0.1% cresyl violet. Images of stained sections were taken and the cerebellar surface area on each section traced and measured by ImageJ. The total volume of each cerebellar region was calculated by multiplying the total surface area of all sections from the same region by the thickness of the sections. Vermis sections were selected as the most medial sections with clearly visible 10 lobules; paravermis sections were adjacent to the vermis sections, with diminishing lobules VIII, IX and X; sections lateral to paravermis were hemisphere sections.

## Whole-mount in situ hybridisation

Noon on the day a vaginal plug was observed was defined as embryonic day (E)0.5. Somite-stage embryos were staged more accurately by counting the number of somite pairs. After dissection in

ice-cold PBS, embryos were fixed overnight in 4% PFA at 4°C, gradually dehydrated in a methanol series and in situ hybridisation carried out using standard procedures (*Wilkinson et al., 1989b*). Digoxigenin-labelled antisense probes for *Etv5* (*Hippenmeyer et al., 2002*), *Fgf8* (*Crossley and Martin, 1995*), *Gbx2* (*Wassarman et al., 1997*), *Hoxa2* (*Wilkinson et al., 1989a*) and *Otx2* (*Simeone et al., 1993*) were prepared using previously published constructs.

## Quantitative RT-PCR analyses

Total RNA was extracted from the mes/r1 region of at least three E9.5 embryos of each genotype using Trizol (Invitrogen, UK) with the addition of 20 µg Ultrapure Glycogen (Life Technologies, UK). A total of 200 ng of RNA was used for first-strand DNA synthesis with the nanoScript Precision RT kit (PrimerDesign Ltd., UK) using random hexamer primers. cDNA synthesis reactions without reverse transcriptase enzyme (no RT) were used as controls for quantitative RT-PCR. Quantitative RT-PCR was performed on a Rotor-Gene Q (Qiagen) using Precision qPCR MasterMix kit with SYBR green (Primerdesign Ltd., UK). All reactions were performed in triplicate. Cq threshold values were determined manually and all were at least 5 Cq values below no RT controls. The Cq values for each sample was normalised to the internal control gene *Ywhaz* (primers provided by Primerdesign) to give the ΔCq value. ΔΔCq values were calculated relative to wildtype samples. The primer sequences used were: *Fgf8*: forward 5′-AGGTCTCTACATCTGCATGAAC-3′, reverse 5′-TGTTCTCCAGCACGATCTCT-3′; *Etv5*: forward 5′-GCAGTTTGTCCCAGATTTTCA-3′, reverse 5′-GCAGCTCCCGTTTGATCTT-3′.

## Chromatin immunoprecipitation (ChIP)-qPCR

The embryonic mes/r1 region was dissected from E9.5 CD1 embryos, disrupted by trituration with a 23 G and 25 G needle, fixed for 10 min with 4% PFA, snap-frozen and stored at −80°C until use. After cell lysis and isolation of nuclei, samples were sonicated in a Bioruptor UCD-300 in 10 mM TRIS pH8, 1 mM EDTA, 0.5 mM EGTA, 0.5% N-lauroylsarcosine to 200–500 bp fragment size. Chromatin was immunoprecipiated with antiserum to CHD7 (ab31824, Abcam, UK) and control Ig (Abcam, UK). Complexes were captured with Protein G Dynabeads, washed with modified RIPA buffer (50 mM HEPES pH7.5, 1 mM EDTA, 0.3% Sodium deoxycholate, 1% NP40, 250 mM LiCl), eluted in 50 mM TRIS pH8, 10 mM EDTA, 1% SDS, cross-links reversed by overnight incubation at 65°C and DNA precipitated after phenol-chloroform extraction. Unique DNA fragments were amplified and quantified by qPCR using the primers in *Tables 2 and 3*. Data were quantified relative to input DNA (% of input).

## Patients

All patients included in this study are known at the Dutch expert clinic for CHARGE syndrome located at the University Medical Center Groningen (coordinated by CvRA). All patients had a pathogenic mutation in *CHD7* (*Table 1*, see also www.CHD7.org). Patients were all evaluated in person by CvRA. Patients and/or parents gave written consent for the collection and analysis of detailed phenotypic information according to national ethical guidelines. Phenotypic information collected also included radiological images. All information is stored in a secure database under a unique patient identification number.

**Table 2.** *Otx2* qPCR primers

| Region | Forward | Reverse |
| --- | --- | --- |
| #1 | AAACTCACCATAATCCTCCTGCC | TCCTCCCCTTCTCCTCTAAACAGC |
| #2 | CTGCTCTCCTCAACCTTCAGACTC | TTGCGTGCCTTACCTTACCG |
| #3 | CAACCACTCAAGTCAAGCCTATCTG | TCTTCCTCTGCCTCCCAAGTTC |
| #4 | CTGGCTGGTGGCTTCTGATT | TTAGGTATCGCCAGGTTGCC |
| #5 | ACACCAACTTGCTGAACAACA | TCCAGACTACTAATTAGGTGAAAATGA |
| #6 | GAAAACCAAAACCCAAACCACG | GAATGGAATCCTTAGCAAGCGG |
| #7 | AACAGGCTTGTGTCCGTCTACG | CGCTTTCTCAGCAAATCTCCC |
| #8 | CATTTTCTTGCCGTCCTGCC | AAAGTGTGCCTCCTGTGGTTCC |
| #9 | AAAAACACTGGGGAAGAAAGGG | AAATAAGAGTCAGAAGAGCGGTGC |
| #10 | GCTGAATCAAACATGAATGAGCC | CTGGGGAGTAGACAACTGAGACA |

**Table 3.** *Gbx2* qPCR primers

| Region | Forward (5′–3′) | Reverse (5′–3′) |
|---|---|---|
| #1 | CCCTTGGCTGGCTTTGAAAT | TCTGCCTTTTGTCCTGGAGA |
| #2 | TGAATCCATAGCTTACCCGC | AGGAACAAAGGGGGAAAGAA |
| #3 | CCAGGCTTTCATCTCTCGCA | ATAGGCCAAGCTAAGCACCC |
| #4 | GGGAATGGTGGAATGAATGGC | TGAGGAGTGTGCTGAAGGGACAAC |
| #5 | GTTGGCTGCCCTTTTCTTCA | ACCTCCATCTCCTCAGGCTA |
| #6 | TGTAAACACTCCCTTCCCCGTATC | CCACCCTAAACCGAAATGCG |

## MRI scans

The MRI scans were made at different hospitals. A total of 23 MRI scans could be collected. Only MRI scans that allowed a reliable interpretation of the cerebellum, that is the presence of sagittal and axial images of the cerebellum, were included in this study (n = 20). All cerebellar images were evaluated on visual basis by an experienced neuroradiologist (LCM). MRI images of CHARGE patients were compared with images of age-matched controls. All observations were recorded by MTYW.

## Acknowledgements

We thank the individuals with CHARGE syndrome and their families for participating in our research. Lies Hoefsloot supervised the diagnostic *CHD7* sequencing of the CHARGE patients. Jorieke van Kammen-Bergman assisted in the collection of cerebral MRI scans. Samantha Martin provided technical assistance and mouse husbandry. We thank Wee-Wei Tee and Shuzo Kaneko for advice on ChIP experiments, Gail Martin, Alex Joyner and John Rubenstein for mouse lines and in situ probes and members of the Basson laboratory and Anthony Graham, Clemens Kiecker and Mohi Ahmed for comments on the manuscript. This work is supported by grants from the Wellcome Trust (091475) and the Medical Research Council (MR/K022377/1) to MAB, Fund Nuts-Ohra (1202–023) to MTYW, and BHF grants PG/12/44/29658 and RG/15/13/28570 to PJS.

## Additional information

### Competing interests

DR: Reviewing editor, *eLife*. The other authors declare that no competing interests exist.

### Funding

| Funder | Grant reference number | Author |
|---|---|---|
| Wellcome Trust | 091475 | M Albert Basson |
| Medical Research Council | MR/K022377/1 | M Albert Basson |
| Fund Nuts-Ohra | 1202-023 | Monica TY Wong |
| British Heart Foundation | PG/12/44/29658, RG/15/13/28570 | Peter J Scambler |
| Howard Hughes Medical Institute | | Danny Reinberg |

The funders had no role in study design, data collection and interpretation, or the decision to submit the work for publication.

### Author contributions

TY, LCM, MTYW, Acquisition of data, Analysis and interpretation of data, Drafting or revising the article; KD, Acquisition of data, Analysis and interpretation of data; TB, Drafting or revising the article, Contributed unpublished essential data or reagents; DR, Analysis and interpretation of data, Drafting or revising the article; PJS, Analysis and interpretation of data, Drafting or revising the article, Contributed unpublished essential data or reagents; CMAR-A, Conception and design, Analysis and

interpretation of data, Drafting or revising the article; MAB, Conception and design, Acquisition of data, Analysis and interpretation of data, Drafting or revising the article

## Ethics

Human subjects: Research involving human subjects was carried out in accordance with the Declaration of Helsinki. Informed parental consent was obtained for the anonymous use of the molecular, clinical, and neuroradiological data reported in this manuscript. The original study was registered at the University of Groningen (2007/211) and supported by a grant of the The Netherlands Organisation for Health Research and Development (ZonMW 92003460) until 2010.

Animal experimentation: All animal procedures were approved by the UK Home Office (PPL 70/7506).

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
