## [Decision Letter]

Thank you for sending your work entitled “Deregulated FGF and homeotic gene expression underlies cerebellar vermis hypoplasia in CHARGE syndrome” for consideration at *eLife*. Your article has been favorably evaluated by a Senior editor and 4 reviewers, one of whom is a member of our Board of Reviewing Editors.

The following individuals responsible for the peer review of your submission have agreed to reveal their identity: Robb Krumlauf (Reviewing editor); Kathleen Millen and Mark Lewandowski (peer reviewers).

The Reviewing editor and the other reviewers discussed their comments before we reached this decision, and the Reviewing editor has assembled the following comments to help you prepare a revised submission.

There was a uniform opinion among the reviewers that this was an interesting paper thath would be suitable for publication in *eLife* following revisions to address several concerns.

1) This study formally adds cerebellar malformation to the features of CHARGE, demonstrating that isthmic disruption can cause human cerebellar malformations as has long been demonstrated in mice and other models. However, the authors oversell the connection by claiming (in the Abstract and start of the section on cerebellar defects) that their mouse observations “led us to search for cerebellar defects...”. Rather, several previous clinical reports clearly described the occurrence of vermis deficiency as a non-random association with CHARGE, although the authors' series appears the largest in which an MRI study has been conducted.

2) An additional major concern is with the phenotypic features of the MRI data in Table 1. As Barkovich and others have pointed out, Dandy-Walker Variant is not acceptable terminology (see more recent references Barkovivh 2009; Aldinger 2009). Dandy-Walker malformation is a combination of cerebellar hypoplasia and posterior fossa enlargement. These MRI scans show cerebellar vermis hypoplasia. There is not clear evidence of an enlarged posterior fossa.

3) In the cases shown, the enlarged fourth ventricle is simply filling a normal-sized or perhaps even small posterior fossa. It is an illusion that the cisterna is enlarged. The cerebellum is small, so the space looks large. Other features are noted, but are not visible in the panels presented – in particular the large foramen of Magendi and indented dorsal pons, abnormal foliation in 2 cases. These issues in establishing the connections of cerebellar defects to CHARGE and the interpretation need to be properly dealt with in the text.

4) An important aspect of the work relates to the conclusion that *Chd7* directs binds regulatory regions of the *Otx2* locus. However there are concerns about the technical aspects of this data. The ChIP-PCR study of *Otx2* in Figure 4 is weak and heavily influenced by a genome-wide ChIP-seq study of murine ES cells described by Schnetz et al. The authors (1) did not provide primer sequences for the regions of *Otx2* analysed (which were chosen from the Schnetz work) and more importantly (2) did not provide any negative control data, for example comparing with other regions of the *Otx2* gene sequence. Although there are some suggestive correlations with other epigenetic marks, there are too few analyses to contextualise the signal they obtained and therefore to interpret it, as they do, as “suggesting a role for CHD7 in directly regulating *Otx2* gene expression through interacting with these elements”. This part of the work would ideally need to be strengthened by conducting a complete PCR-based survey of the *Otx2* gene. Even then, since CHD7 may not bind DNA directly, the conclusions are indirect without a lot more information on co-binding factors. Since this is likely to be beyond the scope of this work, the authors should revise the text to interpret the direct action of *Chd7* on *Otx2* more circumspectly, taking into consideration these potential caveats.

5) Did they consider undertaking a similar study of *Gbx2*?

6) The data in Figure 3 do not clearly show differences in *Gbx2* and *Otx2* expression. While it is clear at the later stages the data at E7.5 are not very compelling and add little to the story. They should either be removed, or to clarify the embryonic data at E7.5 the authors should repeat the *Otx2/Gbx2* analysis with embryos that are more closely stage-matched. In particular, the *Otx2*-stained normal embryo is clearly older than the *Chd7* mutant.

7) The posterior expansion of *Otx2* is an important observation. Figure 3 shows a 6ss embryo doubly stained for both *Otx2* and *Hoxa2*. The expanded region of label is interpreted to be due to caudally expanded *Otx2* expression, but it could also be due to *Hoxa2* expression rostrally expanded from r2. This could be resolved with two color “in situs”. However, another solution is to stain solely for *Otx2* and place a photo of such a sample as an insert in 2D. Alternatively a more compelling way to illustrate the shifts and a potential transformation would be to employ *Krox20*. It is expressed in r3 and r5. This would indicate that the changes were restricted to r1 and not a truncation of the anterior hindbrain. This general issue needs to be addressed in one of the ways suggested above.

---

## [Author Response]

*1) This study formally adds cerebellar malformation to the features of CHARGE, demonstrating that isthmic disruption can cause human cerebellar malformations as has long been demonstrated in mice and other models. However, the authors oversell the connection by claiming (in the Abstract and start of the section on cerebellar defects) that their mouse observations “led us to search for cerebellar defects...”. Rather, several previous clinical reports clearly described the occurrence of vermis deficiency as a non-random association with CHARGE, although the authors' series appears the largest in which an MRI study has been conducted*.

We have revised the text to clarify our findings in the context of previous reports on cerebellar vermis defects observed in pre-term CHARGE fetuses to avoid giving the impression that we are over-selling our clinical findings.

In the Abstract we simply state that: “Finally, we report cerebellar vermis hypoplasia in 35% of CHARGE syndrome patients with a proven *CHD7* mutation.”

And in the text: “Cerebellar defects have been reported in pre-term CHARGE fetuses (4; 34; 25). To determine whether cerebellar defects are a common post-natal feature of CHARGE syndrome, we systematically examined cerebellar structure in a cohort of 20 patients with CHARGE syndrome and mutations in the *CHD7* gene.”

*2) An additional major concern is with the phenotypic features of the MRI data in*
Table 1*. As Barkovich and others have pointed out, Dandy-Walker Variant is not acceptable terminology (see more recent references Barkovivh 2009; Aldinger 2009). Dandy-Walker malformation is a combination of cerebellar hypoplasia and posterior fossa enlargement. These MRI scans show cerebellar vermis hypoplasia. There is not clear evidence of an enlarged posterior fossa*.

We thank the reviewers for pointing these out. We have changed the description by removing references to DWV and posterior fossa enlargement. We amended the text to clarify the relation between the observed cerebellar defects and Dandy-Walker malformations: “Thus, cerebellar defects in CHARGE syndrome have some clinical similarities to Dandy-Walker malformations (vermis hypoplasia and anticlockwise rotated vermis), without the overt posterior fossa enlargement typical of Dandy-Walker malformation (10).”

*3) In the cases shown, the enlarged fourth ventricle is simply filling a normal-sized or perhaps even small posterior fossa. It is an illusion that the cisterna is enlarged. The cerebellum is small, so the space looks large. Other features are noted, but are not visible in the panels presented – in particular the large foramen of Magendi and indented dorsal pons, abnormal foliation in 2 cases. These issues in establishing the connections of cerebellar defects to CHARGE and the interpretation need to be properly dealt with in the text*.

We have amended the text to clarify that the large appearance of the foramen of Magendi and fourth ventricle and large subcerebellar cistern are as a consequence of vermis hypoplasia, making these spaces appear larger: “As a consequence of these abnormalities, fluid-filled spaces surrounding the cerebellum appeared larger. Examples of large foramen of Magendi and fourth ventricle (50%, 10/20) and large subcerebellar cistern (25%, 4/20) are indicated in Figure 5.”

*4) An important aspect of the work relates to the conclusion that* Chd7 *directs binds regulatory regions of the* Otx2 *locus. However there are concerns about the technical aspects of this data. The ChIP-PCR study of* Otx2 *in*
Figure 4
*is weak and heavily influenced by a genome-wide ChIP-seq study of murine ES cells described by Schnetz et al. The authors (1) did not provide primer sequences for the regions of* Otx2 *analysed (which were chosen from the Schnetz work) and more importantly (2) did not provide any negative control data, for example comparing with other regions of the* Otx2 *gene sequence. Although there are some suggestive correlations with other epigenetic marks, there are too few analyses to contextualise the signal they obtained and therefore to interpret it, as they do, as “suggesting a role for CHD7 in directly regulating* Otx2 *gene expression through interacting with these elements”. This part of the work would ideally need to be strengthened by conducting a complete PCR-based survey of the* Otx2 *gene. Even then, since CHD7 may not bind DNA directly, the conclusions are indirect without a lot more information on co-binding factors. Since this is likely to be beyond the scope of this work, the authors should revise the text to interpret the direct action of* Chd7 *on* Otx2 *more circumspectly, taking into consideration these potential caveats*.

We thank the reviewers for these criticisms, which have allowed us to improve our analysis of CHD7 association with Otx2 regulatory elements significantly. First, we performed an unbiased screen of the Otx2 gene that led to the identification of several regions that show specific CHD7 association, as well as adjacent negative control regions that show no CHD7 binding. Secondly, instead of being led by the Schnetz ES cell data, we tested CHD7 binding around known regulatory regions identified by the Aizawa group to drive Otx2 expression in the embryonic fore- and midbrain after E9.5, referred to as the FM1 and FM2 enhancer regions, as well as an early epiblast/anterior neuroectoderm (AN) enhancer region. We could demonstrate strong, specific association of CHD7 with these regions. All primer sequences used have been included in the revised manuscript.

We have revised the text to clarify that the association of CHD7 with Otx2 and Gbx2 (see below) regulatory elements do not constitute incontrovertible proof that CHD7 directly regulates these genes. We also include the caveat that the mechanisms whereby CHD7 might be recruited to these regions and the actual role of CHD7 at these elements are not known and fall beyond the scope of the present study.

*5) Did they consider undertaking a similar study of* Gbx2?

Given the success of our Otx2 ChIP experiments, we also screened the Gbx2 gene, with emphasis on the 10kb region upstream of the gene, as this region likely encompasses an enhancer region that can direct Gbx2 expression to r1, based on studies in zebrafish. We could demonstrate specific CHD7 association with regions upstream of Gbx2. In addition, some of the elements tested represented negative controls with no CHD7 binding. These data are included in the revised manuscript.

*6) The data in*
Figure 3
*do not clearly show differences in* Gbx2 *and* Otx2 *expression. While it is clear at the later stages the data at E7.5 are not very compelling and add little to the story. They should either be removed, or to clarify the embryonic data at E7.5 the authors should repeat the* Otx2/Gbx2 *analysis with embryos that are more closely stage-matched. In particular, the* Otx2*-stained normal embryo is clearly older than the* Chd7 *mutant*.

We agree with the reviewers that these data do not add much to the story, in particular since the Otx2 enhancers that drive expression at E7.5 are likely different from those that control expression in the mes/r1 region at the stages we deal with in the manuscript. We have therefore removed these data and included additional analyses of E8.25 embryos as requested.

*7) The posterior expansion of* Otx2 *is an important observation.*
Figure 3
*shows a 6ss embryo doubly stained for both* Otx2 *and* Hoxa2*. The expanded region of label is interpreted to be due to caudally expanded* Otx2 *expression, but it could also be due to* Hoxa2 *expression rostrally expanded from r2. This could be resolved with two color “in situs”. However, another solution is to stain solely for* Otx2 *and place a photo of such a sample as an insert in 2D. Alternatively a more compelling way to illustrate the shifts and a potential transformation would be to employ* Krox20*. It is expressed in r3 and r5. This would indicate that the changes were restricted to r1 and not a truncation of the anterior hindbrain. This general issue needs to be addressed in one of the ways suggested above*.

We thank the reviewers for these constructive criticisms. We addressed this point in two of the ways suggested. We include pictures of 4ss embryos hybridized with only Otx2 that clearly shows the upregulated, expansion of Otx2 expression. Additionally, we repeated the analysis using Krox20 as a marker for r3+r5 to clearly demonstrate the posterior expansion of Otx2 expression and lack of posterior hindbrain truncation.